# Treatment of Food Aversion and Eating Problems in Children with Short Bowel Syndrome: A Systematic Review

**DOI:** 10.3390/children9101582

**Published:** 2022-10-19

**Authors:** Francesca Gigola, Virginia Carletti, Riccardo Coletta, Martina Certini, Marco Del Riccio, Caterina Bortolotti, Antonino Morabito

**Affiliations:** 1Department of Neuroscience, Psychology, Drug Research and Child Health (NEUROFARBA), University of Florence, 50139 Florence, Italy; 2Department of Paediatric Surgery, Meyer Children’s Hospital, 50139 Florence, Italy; 3School of Environment and Life Science, University of Salford, Salford M5 4NT, UK; 4Postgraduate Medical School of Hygiene and Preventive Medicine, University of Florence, 50134 Florence, Italy; 5Faculty of Medicine and Surgery, University of Florence, 50134 Florence, Italy

**Keywords:** eating disorders, intestinal failure, enteral feeding, parenteral nutrition, mouth, paediatric patients, oral feeding intolerance

## Abstract

Food Aversion (FA) is a strong refusing behaviour to the oral assumption of food that can affect children with Short Bowel Syndrome (SBS). Management includes behavioural and Messy Play treatments, with few reports on systematic strategies to return the patient to enjoyable eating. We conducted a systematic review to better understand this complex and vital issue. (1) Materials and Methods: We investigated publications using MEDLINE, Embase, and the Web of Science to include articles published up to July 2022. The inclusion criteria were original articles including paediatric patients (aged < 18 years old) affected by SBS and Intestinal Failure (IF) who underwent treatment for FA. (2) Results: A total of 24 patients received treatment—15 (62.5%) patients were male and 9 (37.5%) were female. The age range was from 1 month to 16 years. Treatment of FA was carried out by behavioural therapy in 2 patients and Messy Play Therapy in 12 patients already surgically and pharmacologically managed for SBS. The treatment results showed complete weaning from Parenteral Nutrition in 9/14 cases (64%) using the behavioural treatment and 7/12 cases using Messy Play Therapy. (3) Conclusions: FA is a rare but disabling condition that often affects SBS patients, worsening their overall health and quality of life. This condition should be addressed in an Intestinal Rehabilitation Centre context. Our review sheds light on the literature gap regarding FA, and further studies are required to understand better which treatment options best suit SBS paediatric patients.

## 1. Introduction

Food Aversion (FA) is defined as a strong, adverse reaction to the oral consumption of food with variable behavioural responses, such as vomiting, coughing, crying, and refusing to introduce food by mouth. Aspiration and even fatal airway obstruction are possible because of the reactions to food offerings [1]. FA can be experienced both by children and adults, and in some cases, it can be restricted to only certain types of foods or textures. Dietary selectivity among children is common and it is often based on food texture or smell [2]. Around 50% of children show selective eating in early childhood, but many of them will outgrow this behaviour later in life. When food aversion is present, though, it can impact the child’s growth and create conflict and anxiety regarding mealtimes [3,4].

Oral nutrition is of the utmost importance in patients affected by Short Bowel Syndrome (SBS), especially in the first period of life.

For these reasons, children suffer from lengthy hospitalisations and are exposed to many stressful stimuli concerning the oral cavity. They are nutritionally supported by Parenteral Nutrition (PN) and nasogastric tubes or gastrostomy for long periods, exerting an essential impact on their ability to develop swallowing and suction skills [5].

Comorbidities associated with SBS, or SBS itself, may determine the delayed introduction of food by mouth in affected children; this could impact later food choices and feeding development [5]. Suppose an infant who experiences negative stimuli in the oral cavity in the first phase of life, such as tube insertion, oral hygiene, and suctioning. In that case, the child could identify oral feeding with stressful experiences, possibly leading to FA [6]. In 1964, Illingworth et al. [7] postulated the existence of “critical windows”, specific periods in which children should be offered solid food to develop oral skills. The concept of critical periods had been demonstrated in animals before, and it could be associated with personality and environmental factors. If children are not offered solid food when they learn to chew, they could develop difficulties introducing these foods later on [7]. To avoid FA, oral nutrition should be started as soon as possible with breast milk or formula, and critical windows can be exploited to help the child to develop feeding skills. The presence of selective eating and FA in children could also create distress within the home environment and push families to adopt force-feeding to ensure the consumption of proper food quantities [8]. Jansen et al. found a positive correlation between pressure to eat and fussy eating and a bi-directional association between food refusal and forcing behaviours in parents [9]. Parents could also try to convince their children to eat by offering rewards after the consumption of non-preferred foods or new food. This could impact the child’s ability to understand his or her preferences through experience and could lead to negative consequences [10]. Milano et al. [11] recently proposed a stepwise approach to the treatment of feeding difficulties in children, defining short-term goals while also considering long-term desired results. The first step should be the identification of life-threatening conditions, such as aspiration and growth failure, that should be addressed. The involvement of families is mandatory, and parents should be given guidelines on how to manage mealtimes, such as avoiding distractions, limiting meal duration, and offering new foods without pressure to eat [11].

The prevalence and treatment of FA in SBS are not well studied in the literature, with few reports on systematic strategies to return the patient to enjoyable eating. Treatment strategies in SBS patients should also take into consideration the comorbidities of these children and the reliance on PN for most of them. What we do know is that the treatment of FA is a multidisciplinary topic involving many different professional figures, such as physicians, nurses, dietitians, social workers, occupational therapists, psychologists, and physical therapists [6]. Possible treatments vary depending also on the patient’s characteristics. Preventative treatment is usually used in non-orally fed children that have reached clinical stability and can tolerate oral stimuli. If the patient is a child who has already experienced oral feeding, a technique called oral sensorimotor skill-building can be employed; if the child does not show signs of a desire to eat, attempts could be made with hunger provocation. Eventually, if the patient has comorbidities, such as being on the autism spectrum, sensory integration therapy can be a possibility [12].

We conducted a systematic review on this topic to better understand the complex and important issue of FA and whether a structured protocol could be suggested.

## 2. Materials and Methods

### 2.1. Protocol and Search Strategy

This systematic review was conducted according to the Preferred Reporting Items for Systematic Reviews and Meta-Analyses (PRISMA) statement [13] and the protocol of this study was registered prior to the beginning of the investigation in the International Prospective Register of Systematic Reviews Database (PROSPERO) (registration number: CRD42022329838).

We conducted a literature search using MEDLINE, Embase, and the Web of Science to include articles up to July 2022 using the following keywords: food aversion, oral aversion, feeding intolerance, occupational therapy, eating behaviour, intestinal failure, and short bowel syndrome. No language, time, or geographical restrictions were applied as long as an English abstract was available to decide on eligibility.

The inclusion criteria were original articles including paediatric patients (aged < 18 years old) affected by SBS and IF who underwent treatment for FA. The primary endpoint was the efficacy of therapy in improving oral feeding. The parameters considered were increased oral intake, weaning from PN or decreased PN dependency, ability to taste all consistencies of food, and increased amount of oral water in the total water requirement.

### 2.2. Study Selection and Data Extraction

After excluding duplicates, articles were screened for inclusion by title and abstract by three authors (F.G, V.C., and M.C.) independently in a blind manner. Any disagreement was resolved by consensus.

Data were extracted using an internal spreadsheet, and the following information was extracted: (1) Study characteristics—title, first author, year of publication, country of the hospital in which the study was conducted, study design, number of patients enrolled, and number of patients treated; (2) characteristics of participants and treatment; and (3) outcome of treatment.

Given the paucity of available studies and patients identified with our selection criteria, data are reported in a narrative review. Study-specific results are reported in tables and commented on in the text without making any attempt to obtain summary measures of association.

Two authors performed a quality assessment of the studies independently and blindly, according to the JBI Critical Appraisal Tools for Case Reports and Case Series [14]. Any disagreement was resolved by consensus. The results of the quality assessment are available in the Appendix A.

## 3. Results

A total of 1838 articles were identified from the literature search (503 from MEDLINE, 886 from EMBASE, and 449 from Web of Science) (Figure 1).

After removing 410 duplicates, 1428 studies were screened for inclusion based on title and abstract review. After full-text selection, three studies fulfilled the inclusion criteria and were finally included in our review. The main reasons for exclusion were the type of study (systematic review and meta-analysis), study population over 18 years old, and study population suffering from other gastrointestinal diseases or having an intact bowel.

The main characteristics of the included studies are summarised in Table 1.

All included studies provided clear information regarding the patients’ demographic and clinical conditions. Both case reports provided a clear and thorough definition of the treatment program and its results, post-intervention clinical conditions, and extended follow-up [15,16]. The case series clearly described the inclusion criteria and demographics of all included patients, as well as clinical information and underlying conditions. This study also provided a definition of FA, and the results were presented thoroughly with information on follow-up and treatment discontinuation. Overall, the included studies have good methodological quality with a low risk of bias.

A total of 14 patients received treatment—6 (50%) patients were male and 6 (50%) were female. The characteristics of the included patients are listed in Table 2. The age range was from 4 years to 16 years. All included studies were single-centre studies. All patients were affected by SBS and FA, defined as redding intolerance (liquids and solid foods refusal, vomiting, regurgitation pain, and bloating associated with enteral feeds).

The causes of SBS were gastroschisis (6), intestinal atresia (2), necrotising enterocolitis (3), volvulus (2), malrotation (1), and Hirschsprung’s disease (1). Table 3 summarises the causes of intestinal resections.

Treatment of FA was carried out by behavioural therapy in 2 patients and Messy Play Therapy in 12 patients. The duration of treatment ranged from 37 days to 10.5 months. The treatment results showed complete weaning from Parenteral Nutrition in 2 cases out of 2 (100%) using Behavioural treatment and 7 out of 12 (58%) using Messy Play Therapy. Table 4 summarises information regarding treatment.

SBS is a complex and multi-systemic disease, and children affected by SBS are usually managed within Intestinal Rehabilitation Programs (IRPs). Treatments for SBS include pharmacological therapies and surgery, such as bowel-lengthening procedures and bowel transplantation. The final goal of treatment is to achieve enteral autonomy and wean from parenteral nutrition. Treatment of FA itself is not sufficient to achieve enteral autonomy—the selected patients were already surgically and pharmacologically managed for SBS.

Behavioural treatment was not performed with a unanimous pathway in the studies analysed. Linsheid et al. [15] admitted the patient for 20 days during the feeding program. Mealtime was limited to 25 min, the feeder was at a one-meter distance from the patient, and the patient would receive rewards and praise after ingesting the goal quantity for each meal; otherwise, those rewards were lost for 2 h. At admission, only nocturnal gastrostomy tube-feeding was maintained to reduce the patient’s awareness of artificial nutritional intake. Meals were provided in a distraction-free environment, and various liquid, semisolid, and solid foods were introduced sequentially and gradually increasing in quantity. When the patient was introduced to a predetermined amount of a particular liquid or solid food, a new one was introduced. During inpatient treatment, the patient’s mother observed some of the sessions, and upon discharge, she was instructed on how to continue the treatment at home. At the beginning of each meal, the patient was shown the eating goal for that meal: if the target was achieved, the patient was praised; if not, he received a short time-out in his room. At a six-month follow-up, all food and liquid intake were by mouth, and the patient had expanded the range of foods accepted.

Groff et al. [16] admitted the patient for a day treatment program; the patient refused solids and fluids, and the initial treatment increased the consumption of solids on a spoon, but not liquids from a cup. The mealtime was 45 min, and the authors used spoon-to-cup fading to increase the consumption of drinks from a cup. Similarly to Linsheid et al. [15], the observer sat at 1.5 m from the patient, and the patient received praise in case of complete swallowing. There was no punishment in case of refusal. Fading consisted of altering the spoon by retaping the spoon’s bowl closer to the edge of the cup. Caregivers were instructed on how to provide the treatment at home, and the patient was followed up for one year. At that time, the patient could accept drinks from a cup with a 100% acceptance rate.

Chiatto et al. [17] retrospectively analysed data on children with SBS suffering from FA and treated with a technique called Messy Play Therapy (MPT). The aim of MPT is to allow children to become familiar with food textures and smells without pressure to taste or eat it. Sessions lasted for 30 min, 2–3 times a week; as in-patients, sessions were held in the department; when discharged, they were conducted at home with specialists. The average duration of therapy was 10 months. The program began with desensitisation to touch using different materials, and afterwards, foods were introduced and used in fun activities. Eventually, the child would experience different tastes, and caregivers were actively involved in sessions. All patients tolerated liquids before MPT. Before treatment, 41.7% of patients did not accept pureed food, and afterwards, all children took purees with a difference that was not statistically significant. Tolerance to taste was also affected and improved after therapy. There was a statistically significant increase in tolerance to savoury and sweet food after therapy. At the end of the treatment period, seven patients were weaned from PN.

## 4. Discussion

FA in SBS patients appears to be poorly represented in the recent literature, with very few findings on how to treat such an important issue. Epidemiologic data on the prevalence of FA in the paediatric population are scarce, and data collection is impaired by the possibility that symptoms of FA may be disguised as symptoms of the disease (such as dysphagia or retching) and the lack of standardised protocols and definitions. Linscheid [18] and Burklow et al. [19] estimated the prevalence of feeding problems in the paediatric population as ranging from 25 to 45% in typically developing children and 33 to 80% in children with developmental delays. In 2019, Goday et al. [20] defined Paediatric Feeding Disorders (PFD), including problems associated with medical conditions and comorbidities; they defined PFD as “impaired oral intake that is not age appropriate and is associated with medical, nutrition, skill, or psychosocial dysfunction”. SBS patients are at high risk of developing PFD and FA due to their underlying condition and the inability, in most cases, to sustain growth and absorb fluid and nutrients enterally. In these patients, oral feeding should be offered, even if it does not provide nutritional value. It should be administered even in neonates, when possible, to grant the positive effect of stimulating gut hormones and growth factor production, developing suction and deglutition skills, and avoiding FA. Achieving enteral autonomy in SBS patients is crucial in their treatment, and oral nutrition has a positive effect, enhancing the natural adaptation process of the intestine and the production of growth factors [21,22]. The possibility to eat by mouth is also an important psychological factor for SBS patients and their families, improving their quality of life [23,24]. Weaning patients from PN means reducing the risk of Catheter-Associated Bloodstream Infections (CABSIs), Intestinal Failure, Associated Liver Disease (IFALD), and thrombosis [25,26]. Enteral autonomy can be achieved through different means, i.e., surgical and medical. Among surgical procedures are lengthening and slowing transit procedures [27], while Teduglutide, a GLP2 analogue, has been proven to reduce the need for PN in paediatric patients [28], opening the boundaries of future medical treatments.

For these reasons, preventing and treating FA should be a priority in SBS patients. This condition should include different professional figures conducted in an Intestinal Rehabilitation Centre [29]. Our systematic review has shed light on the gap in the literature regarding FA in SBS patients, with very few reports on the topic with different approaches to the same problem. The paucity of the included studies is a limit of this review, making it difficult to draw specific conclusions regarding treatment efficacy. The definition of FA was homogenous in all selected studies; patients were treated with Behavioural Treatment in two studies and with Messy Play Therapy in one study. Behavioural treatment was, however, conducted in different ways in different studies: one provided nutritional goals associated with rewards and praise if they were met and punishment in case of failure; another study used the fading technique to improve the consumption of liquids from a cup. Furthermore, patients in different studies showed different characteristics regarding the type of FA: some refused both solid and liquid foods, others only refused liquids, some were eating by mouth, while others were primarily fed by gastrostomy or a nasogastric tube.

The included papers showed significant differences in treatment modalities, length of treatment and follow-up, and in terms of the endpoints of efficacy considered. These differences did not allow us to define the precise results of this review, and our work shows the lack of reports on FA in SBS in the literature. Despite the differences found among different papers on how behavioural treatment was conducted, they all showed improvement in food assumption and tolerance of different textures and tastes, with 64% of patients being weaned from Parenteral Nutrition. All the included papers show that gaming, rewards, and distractions are a strong means for children to overcome their fear of oral eating. Children establish a connection to food starting with tactile sensations. This process allows patients to regain confidence in food consistency and in mealtimes again. Furthermore, when using behavioural treatment, all included studies kept parents away during mealtime. This eliminated parental apprehension and a stressful component for children, who may feel frustrated. This action helped to create a more relaxing environment that was also more playful and freer of judgement.

Behavioural treatment is an asset in the context of the multidisciplinary treatment of SBS to approach FA. This approach could contribute to enteral autonomy, enhancing the ability of the child to eat on their own by mouth. Overall, the included studies had good methodological quality with a low risk of bias, but it is important to highlight the fact that only case reports and case series were found in the literature. This should be recognised as another limitation of this review, as case reports and case series are non-analytical studies and they cannot provide the same data quality as case–control studies or even clinical trials.

## 5. Conclusions

FA is a rare but disabling condition that often affects SBS patients, worsening their overall health and quality of life. It should be addressed in a multidisciplinary way in an Intestinal Rehabilitation Centre context. Its treatment is not well-defined and standardised among different centres, with few literature reports regarding this condition. Despite SBS being a potentially devastating disorder, the rarity of this condition has made multicentre studies hard to design; therefore, it is not easy to create definitive pathways. Larger series or multicentre studies would allow investigators to conduct more statistical analyses that can shed light on definitive treatment for FA.

## Figures and Tables

**Figure 1 children-09-01582-f001:**
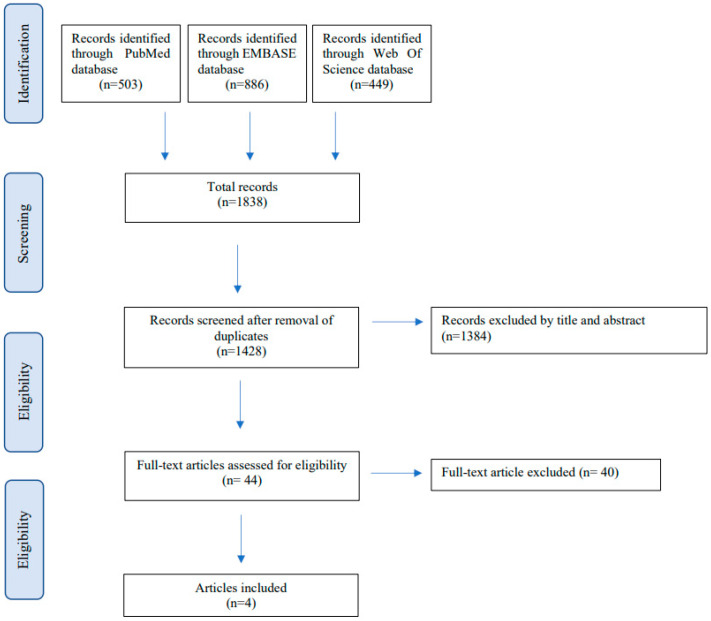
Paper selection according to the PRISMA statement [7].

**Table 1 children-09-01582-t001:** Included studies and main characteristics; all patients enrolled in these studies were treated.

Authors and Year of Publication	Type of Study	Patients Enrolled
Linscheid, 1987 [15]	Case report	1
Groff, 2011 [16]	Case report	1
Chiatto, 2019 [17]	Case series	12

**Table 2 children-09-01582-t002:** Main characteristics of the enrolled patients and related symptoms.

Authors and Year of Publication	Median Age (Years)	Sex (%)	Refuse of Solids and Liquids	Vomiting	Retching	Gagging
Linscheid, 1987 [15]	6	M 100%	Yes	No	No	No
Groff, 2011 [16]	4	M 100%	Yes	No	No	No
Chiatto, 2019 [17]	9	M 33.3%; F 66.6%	Yes	Yes	Yes	Yes

**Table 3 children-09-01582-t003:** Causes of intestinal resection.

Authors and Year of Publication	NEC	Midgut Volvolus	Intestinal Atresia	Gastroschisis	Malrotation	Hirschsprung
Linscheid, 1987 [15]	0	0	1	0	1	0
Groff, 2011 [16]	1	0	0	0	0	0
Chiatto [17]	2	2	1	6	0	1

**Table 4 children-09-01582-t004:** Treatment characteristics and results.

Treatment	No. of Patients	Duration (Mean)	Weaning from PN (%)	Food Tolerance (%)
Behavioural treatment [15,16]	2	37 days	100%	100%
Messy Play Therapy [17]	12	10.5 months	58%	100%

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
