# Peer review of "Treatment of Food Aversion and Eating Problems in Children with Short Bowel Syndrome: A Systematic Review"

_children, 2022, doi:10.3390/children9101582_

Round 1

Reviewer 1 Report

Dear Sirs. 

This paper addresses a very interesting and serious problem in the clinical setting. Albeit rare, the implications are very important in the management of children with SBS. The authors have proposed to perform a systematic review of the literature and could find only 4 previous papers (case or small series reports) that had both the design and the scientific quality to be included in the review. The methods and the selection criteria are well described. The fact that the papers included are not prospective, randomized blinded studies severely impairs any clinicaly usable conclusions but this is addressed in the discussion section. It is believed that the great merit of this study is to stress the importance of food refusal problems in children with SBS. 

Author Response

We thank the Reviewer for this comment. As highlighted, the papers included in our study are not prospective, randomized blinded studies but unfortunately food aversion in SBS is rarely reported in children in literature. Even so we appreciated that the reviewer underlined the importance of our work in stressing the importance of this rare but disabling condition and the necessity of further studies.

Reviewer 2 Report

It is a very interesting topic and finding this gap helps to notice that there should be multi-center studies in this field.

Abstract:

Line 16 I believe that something is missing? Maybe It could say: Management includes pharmacological and behavioural treatments...

Line 26 Treatment results showed complete weaning from Parenteral Nutrition in 9/14 cases (64%) using the behavioural treatment and 2/20 cases (20%) using Cisapride. In this sentence it seems that only the behavioral/pharmacological treatments contributed to wean off PN. You should comment about the Standard of care treatment. The same in Results Line 173-174

Introduction

Line 50 you repeat ability (skills?)

Line 75 If the patient is a child who has already experienced oral feeding, a technique called oral sensorimotor skill building can be employed; while if the child does not show signs of desire to eat, attempts could be done with 78hunger provocation. (punctuation added)

Line 264 In these patients, oral feeding is should be offered even if it does not provide nutritional value.

Line 320 Behavioural treatment is a valuable asset in the context of a multidisciplinary treatment of SBS to approach FA, and it could lead contribute to enteral autonomy. (just to point that it is a contributing factor together with standard of care treatment)

Line 322 Cisapride administration is rarely reported in children in literature...

In Raphael's article, the use of cisapride is proposed for children with difficulties in advancing enteral feeds. The word aversion is not mentioned. Besides most of the patients described have dismotility (gastroschisis, atresia, NEC) and I interpret the symptoms referred are because of dismotility and not because of aversion. I would reconsider excluding this paper from the analysis. 

Perhaps you can mention it in the discussion as a different approach when dismotility is the main cause of feeding intolerance but I believe these patients did not have oral aversion

Author Response

Dear Reviewer

thank you for your precious work and suggestions, we feel the paper was improved by your work. Attached you may find our answers.
